# Synthesis of Green Copper Nanoparticles Using Medicinal Plant *Krameria* sp. Root Extract and Its Applications

**DOI:** 10.3390/molecules28124629

**Published:** 2023-06-08

**Authors:** Shifaa O. Alshammari, Sabry Younis Mahmoud, Eman Saleh Farrag

**Affiliations:** 1Biology Department, College of Science, University of Hafr Al Batin, Hafr Al-Batin 31991, Saudi Arabia; dr.shifaa@uhb.edu.sa; 2Clinical Laboratory Sciences Department, College of Applied Medical Science, University of Hafr Al Batin, Hafr Al-Batin 31991, Saudi Arabia; esfarrag@uhb.edu.sa; 3Microbiology Department, South Valley University, Qena 83523, Egypt

**Keywords:** nanoparticles, copper, antimicrobial, antioxidant, green synthesis, *Krameria* sp.

## Abstract

Nanotechnology is one of the most dynamic research areas and the fastest-growing market. Developing eco-friendly products using available resources to acquire maximum production, better yield, and stability is a great challenge for nanotechnology. In this study, copper nanoparticles (CuNP) were synthesized via the green method using root extract of the medical plant Rhatany (*Krameria* sp.) as a reducing and capping agent and used to investigate the influence of microorganisms. The maximum production of CuNP was noted at 70 °C after 3 h of reaction time. The formation of nanoparticles was confirmed through UV-spectrophotometer, and the product showed an absorbance peak in the 422–430 nm range. The functional groups were observed using the FTIR technique, such as isocyanic acid attached to stabilize the nanoparticles. The spherical nature and average crystal sizes of the particle (6.16 nm) were determined using Transmission Electron Microscopy (TEM), Scanning Electron Microscopy (SEM), and X-ray diffractometer (XRD) analysis. In tests with a few drug-resistant pathogenic bacteria and fungus species, CuNP showed encouraging antimicrobial efficacy. CuNP had a significant antioxidant capacity of 83.81% at 200 g/m^−1^. Green synthesized CuNP are cost-effective and nontoxic and can be applied in agriculture, biomedical, and other fields.

## 1. Introduction

Nanoscience has developed to the level of leading science, with opportunities for fundamental and applied research in all basic cognitive sciences. The emerging field of nanoscience and nanotechnology is driving a global technological revolution [1]. The term is usually applied to materials ranging from 1 to 100 nm. Nanomaterials exhibit different properties than bulk materials due to their size, physical strength, chemical reactivity, electrical conductance, magnetism, and optical effects [2]. Various approaches, such as physical, chemical, and biological, have been developed for the synthesis of controlled size and shape as well as stable nanomaterials [3]. Biological synthesis surpasses the chemical and physical methods in terms of cost, environmental friendliness, and ease of scaling up for large-scale synthesis [4]. The biological method is eco-friendly, economical, safe, and it uses beneficial microorganisms [5]), such as cell cultures of bacteria, fungi, and plants [6]. Plants are preferable to other biological sources for nanoparticle synthesis due to the simple and one-step procedure that avoids the time-consuming process of maintaining cell culture and a non-aseptic environment [7]. Plants are widely available, easy to handle, a source of several metabolites, and rich in pharmacological constituents that act as reducing and capping agents in the synthesis of nanoparticles [8]. Several factors influence the green synthesis method, including temperature, pH, reaction time, metal salt volume, and plant extract volume. The interaction of these factors is critical in determining the shape and size of synthesized NPs [9]. Various plants and their parts, such as the leaf extract of *Hagenia abyssinica* L. [10], the juice of *Citrus medica* Linn. (Idilimbu) [11], and the peel of *Punica granatum* [12] have been reported for the synthesis of CuNP. Rhatany is a hemiparasitic shrub that belongs to the family of Krameriaceae, a monogeneric family of 18 species native to South America [13,14]. Rhatany is also known as (*Krameria lappacea* or *Krameria triandra*) [15]. The therapeutic value of this plant has been reported in the roots [15], specifically in the outer root cortex with dark red periderm [14]. The red root extracts contain mainly tannins (cate-chins and proanthocyanidins) material [14]. The biological activity of Rhatany is due to the existence of the astringent Rhataniatannic acid, which is similar to tannic acid [16]. Rhatany was traditionally used as a remedy for many diseases, such as gastrointestinal diseases, menorrhagia disease, leukorrhea disease, and oral illness [15]. Many medical studies reported the pharmacological and biological properties of Ratany root extracts, including antidiabetic [17], anti-inflammatory [18], antimicrobial [13,15], antioxidant [15], and photoprotective [13,19] effects. CuNP are among the most promising nanomaterial due to their antimicrobial [3] and antiviral properties [20]. These particles are used in many areas, such as in wound healing [21], food packing, and water disinfection [22]. The current study reports the green synthesis of CuNP using the root extract of *Krameria* sp. (Rhatany). It also determines their antimicrobial and antioxidant impact. 

## 2. Results

### 2.1. Synthesis of CuNP

Temperature and time were optimized as reaction parameters. CuNP are formed when the color changes from blue (aqueous salt solution) or brown (pure extract color) to dark brown. Apart from the color transition, the development of a specific absorption peak associated with CuNP at 422–430 nm verifies the green method of nanoparticle synthesis (as seen in Figure 1 and Figure 2). In all the synthetic reactions carried out to optimize the synthesis approach of the CuNP, the characteristic peak in the prescribed reporting range for CuNP was seen (marked in Figure 1 and Figure 2).

### 2.2. Optimization of Conditions for Green Synthesis of CuNP

#### 2.2.1. Effect of Temperature

There was no peak in the root extract solutions incubated at RT, 30 °C, or 40 °C. When the temperature reached 50–70 °C, the synthesis was begun. The solutions extract at 70 °C gave an improved absorption rate (1.98) when compared to 60 °C, and it was selected for further examination. Temperature rise enhanced the production of Cu NPs, and 70 °C was employed for the next process. 

#### 2.2.2. Effect of Time

In various vials, the root extract was combined with 0.3 M copper sulfate solution and stored in an incubator at the desired temperature. After 1 h of response, the lowest absorption rate was observed. At 3 h, the absorption rate of the resultant solutions was higher (1.98) than after 1 (1.74), 2 (1.65), 4 (1.40), 5 (1.37), and 6 h. (0.82). Figure 2 shows a prominent and distinct characteristic peak at 422–430 nm.

#### 2.2.3. Effects of pH

The Rhatany root extracts were reacted with copper sulfate solution in an incubator, and the pH of the solution was adjusted to 5, 7, 9, 11, and 13. No color change was observed at pH 5 and 7, and the UV spectrum showed no characteristic peak. The production of CuNP started with an increase in the pH of the solution up to pH 11. The solution kept at pH 11 gave a broad peak indicating the synthesis of bigger CuNP. The maximum absorption rate (0.441) was shown when the solution mixture and copper sulfate solution were kept at pH 11. 

### 2.3. Physiochemical Characterization of CuNP

To evaluate the absorption spectrum, the purified CuNP extract was dissolved in distilled water and subjected to double-beam BMS-2800 UV-spectrophotometry within 300–800 nm. Figure 2 shows a distinct characteristic peak for CuNP due to the nanoparticles’ surface plasmon resonance phenomena. Figure 2 depicts the UV-visible absorbance spectrum for CuNP, which had a maximum of 426 nm. The X-ray diffractometer (XRD) study of biosynthesized CuNP is shown in Figure 3. It had distinctive diffraction peaks at the (100), (111), and (200) planes, with 2 theta values of 20.4, 36.7, and 48.6°, respectively.

The surface-attached functional groups of biomolecules serving as capping/stabilizing agents on CuNP was examined using FTIR analysis. Peaks in the CuNP’ FTIR spectra were visible at 3351, 2320, 1720, and 1160 cm^−1^ (Figure 4). The broad CuNP absorption peaks at 1741 cm^−1^, which may be inferred from the identified peaks, are indicative of higher isocyanic acid content.

Scanning electron microscopy (SEM) was done on the surface morphology of greenly produced CuNP. According to Figure 5a, a plant extract produced in distilled water was used to create smooth, spherical-to-oval-shaped CuNP nanoparticles that had aggregated and were 5.2 to 7.7 nm sized. It was discovered that the SAED pattern corresponded to particular CuNP crystal planes, as shown in Figure 5b. HRTEM imaging of CuNP is also shown in Figure 5b, at a diameter of roughly 6.16 nm.

### 2.4. CuNP Activity

#### 2.4.1. Antioxidant Assay

Rhatany roots extract, precursor salt (CuSO_4_·5H_2_O), and biosynthesized CuNP was tested for their antioxidant capacity against the DPPH free radical in vitro. Each sample’s ability to scavenge DPPH free radicals in comparison to ascorbic acid was assessed. The obtained results in Figure 6 showed that the DPPH scavenging activity percentage was strongly connected with concentrations and that the activity of the samples against DPPH radical was dose-dependent. The precursor salt has the lowest anti-DPPH activity (42.78 at 200 µg/ml^−1^). Due to the presence of phenolic components, which are certain to have a free radical scavenging nature, the extract demonstrated superior antioxidant activity of 76.02 at the same dose. The maximum DPPH radical scavenging activity was shown by the biosynthesized CuNP (83.81) at 200 µg/ml^−1^ concentrations. Possibly as a result of the higher concentrations, for most of the treatments, there was a significant effect of the different treatments on the DPPH radical scavenging activity (Figure 6). Compared with ascorbic acid, the increase in CuNP concentration led to a significant increase in the DPPH radical scavenging activity from 55 to 83% (*p* > 0.05).

#### 2.4.2. Antimicrobial Activity

Using the Mueller–Hinton agar and disc-diffusion method, the antimicrobial activity of Rhatany root extract and greenly generated CuNP was examined against a variety of bacterial and fungal species. By assessing the zones of inhibition and using chloramphenicol as a positive control, the antibacterial activity of extract and CuNP was assessed against drug-resistant bacteria (Staphylococcus aureus and *Escherichia coli*). Moreover, plant pathogenic fungal strains (*Alternaria alternata* and *Fusarium oxyporium*) were used to test the antifungal activities of extract and CuNP.

##### Antibacterial Activity

As seen in Figure 7, all chosen drug-resistant bacterial strains responded favorably to both extract and green-manufactured copper nanoparticles in vitro. As compared to chloramphenicol (the gold standard medicine), the efficacy of extract and CuNP was proven to be significant in most cases based on their concentrations for two bacterial strains. The highest zone of inhibition was discovered for *S. aureus* (Figure 8). 

The highest doses of Rhatany root extract have shown antibacterial activity against both bacterial strains, with respect to the *S. aureus* and *E. coli* bacterial strains, with MICs of 128 µg/ml. They displayed the largest zone of inhibition (4.7 mm) and the lowest zone of inhibition (1.5 mm) for *E. coli* and *S. aureus*, respectively, by using 128 µg/mL.

The greatest zone of inhibition (43.4 and 44.1 mm) at 256 µg/ml^−1^ of CuNP against *E. coli* and *S. aureus* strains, respectively. Chloramphenicol marginally outperformed CuNP (46.2 mm). With the same concentration (160 µg/mL) against *E. coli* as the other studied substances, it demonstrated the lowest zone of inhibition. Statical analysis indicated significant activity of CuNP against *E. coli* and *S. aureus* strains at 32 µg/ml^−1^. The activity was highly significant by increasing CuNP concentration (Figure 7). 

##### Antifungal Activity

The antifungal activity of Rhatany root extract and biosynthesized CuNP against plant pathogenic fungal strains is introduced in Figure 9. In comparison with fluconazole (positive control), CuNP showed promising and significant antifungal activity, while the extract showed the lowest inhibition zones against both fungal strains. The plant extract in Figure 9 revealed the highest zone of inhibition (13 nm) against *A. alternata* and the lowest zone of inhibition (11 nm) against *F. oxysporum*. However, the biosynthesized CuNP showed inhibitory ability in both fungal and bacterial strains; it demonstrates less activity in fungi when compared to bacteria. 

Statical analysis indicated high significance against *A. alternata* compared with root extract that at lower concentrations (10 and 20 µg/mL^−1^) of CuNP. Whereas there is no significance at 40 µg/mL^−1^. On the other hand, the activity of CuNP was significant against *F. oxysporum*, but was not significant against both fungi by increasing the concentration of CuNP to 160 µg/mL^−1^ (Figure 9).

## 3. Discussion

Biosynthesis of CuNP was performed by using an aqueous extract of Rhatany root. Reaction parameters (incubation time and temperature) were optimized as described in Figure 1. The change in color from blue (aqueous salt solution) and dark brown (pure extract color) to reddish-brown indicates the formation of CuNP. In addition to the color transition, the creation of a characteristic CuNP absorption band at 406–410 nm also supports the nanoparticles’ generation via the green route, as shown by [10,23]. Since the absorbance at 0.3 M (426 nm) generated the sharpest peak, the maximum yield [24] of CuNP was obtained in this concentration. The medium and yield of the reaction can also be influenced by temperature, which is another crucial component. As a result, the influence of temperature on the biosynthesis of CuNP is maximized by changing its value between 25 and 70 °C while maintaining the same values for all other reaction variables. Similar to how other parameters affected the medium’s response characteristics, temperature fluctuation [25]. As a result, the phytochemicals’ capacity to withstand heat was crucial for the biosynthesis of CuNP. The phytochemicals would break down, and the stability and reduction of nanostructures would be suppressed if the proper temperature had not been employed during the synthesis [26]. FT-IR spectrum gives stretching and bending vibrational frequencies of phenolic−OH that are assumed to be involved in the broadband at 3422 cm^−1^. The small peak observed at 2326 cm^−1^ was also attributed due to the stretching and bending CN of cyanide groups. The small peak observed at 1107 cm^−1^ was also attributed due to the stretching and bending NH of amino groups. Copper nanoparticles (CuNP) were found to form round to oval crystals, as indicated by XRD [3]. The two spots on the SAED pattern were found to correspond to specific CuNP crystal planes observed in XRD measurements. Concentric circles were used to indicate the most prominent three planes, which correspond to the (100), (111), and (200) planes, as well as the XRD result. The obtained results showed that the DPPH scavenging activity percentage was strongly connected with concentrations and that the activity of the samples against DPPH radical was dose-dependent. The precursor salt showed the lowest anti-DPPH activity percentage. Due to the inclusion of phenolic components, which are guaranteed to have free radical scavenging properties, the extract demonstrated greater antioxidant activity at the same concentration. The most effective CuNP for scavenging DPPH radicals were those produced by biosynthesis. The inhibitory activity of the CuNP was found to be more advanced than the observed anti-DPPH potency of precursor salt and extract at comparable doses. The bulk of studies indicates that the features of nanoparticles, such as their size, shape, surface area, and charge distribution, can influence how successfully they pass through biological barriers and interact with the cellular environment [27,28]. CuNP made by biosynthesis displayed promising antibacterial activity. This could occur as a result of the unusual physicochemical properties of the generated nanoparticles, such as their small size, vast surface area, and strong charge distributions. These characteristics may also be affected by controlling variables such as precursor salt concentrations, plant extract, pH, temperature, and reaction time. Despite the fact that the precise mechanism underlying CuNP’s antibacterial activity is unknown, some of the most well-liked theories for how they function in general include the adhesion of these particles to bacterial surfaces, penetration inside the cell and destruction of bacterial biomolecules and intracellular structures, production of reactive oxygen species (ROS) and free radicals that cause cellular toxicity and oxidative stress, and modification of the bacterial signal transduction pathway [29]. Although the biosynthesized CuPNs demonstrated inspiring ability in both fungal and bacterial strains, fungi are showing less activity than bacteria. This may be related to fungus having chitin, which is composed of polysaccharides with a nitrogen group and N-acetylglucosamine, and a harder cell wall. As a result, it is difficult for samples to move between the exterior and interior layers of the cell wall. However, the peptidoglycan (a polymer made of sugars and amino acids) found in bacteria’s cell walls is less stiff and makes sample passage easier. 

Over the last few decades, chemical and physical synthesis approaches have been used to create nanoparticles. However, these have been shown to be hazardous to both the environment and human health. Biological synthesis technologies have been identified as potential alternatives. Our findings confirm that these approaches are promising in a variety of contexts, including green chemistry. Copper nanoparticles can be manufactured in large quantities using Rhatany root extract as a “renewable source.” When compared to chemical procedures, it can be utilized in large numbers with no issues.

## 4. Materials and Methods

### 4.1. Materials

*Krameria* sp. (Rhatany) root tissue was purchased from local markets at Saudi Arabia and washed using distilled water several times to remove all the dust. Then, tissue was dried at room temperature under shade, ground with the help of an electric grinder, and stored. *Escherichia coli*, *Staphylococcus aureus*, *Alternaria alternata*, and *Fusarium oxyporium* were brought from previous work at the University of Hafr Al Batin [30]. All the required chemicals, i.e., copper salt (CuSO_4_·5H_2_O) was bought from Sigma–Aldrich through Science Traders.

### 4.2. Preparation of Extract and Reagent

The plant extract was prepared using maceration and boiling methods. Firstly, powdered material of Rhatany root was added to water with a 1:10 (*w*/*v*) ratio. The mixtures were kept overnight at room temperature, followed by filtration through Whatman’s No. 1 filter paper, and the filtrate was used to synthesize CuNP. 0.3 M copper salt (CuSO_4_·5H_2_O), was prepared by mixing the required amount of copper sulfate pentahydrate to appropriate volume of Rhatany root extract.

A mixture of 20 mL of filtered plant extract solution and 80 mL of distilled water was prepared. The mixture was then shaken for 24 h with 10 mM of 0.3 M copper salt (CuSO_4_·5H_2_O). Following incubation, the solution was centrifuged at 10,000 rpm for 10 min. The precipitated particles were rinsed with deionized water. The precipitated pellets were dried in a hot-air oven at 80 °C for six to eight hours before being stored in appropriate containers. The purified CuNP was then characterized.

### 4.3. Optimization of Conditions for Green Synthesis of CuNP

#### 4.3.1. Effect of Temperature

The temperature of the reaction mixture was optimized by adding 10 mL of freshly prepared extracts with 10 mL of 0.3 M CuSO_4_·5H_2_O solution in glass vials. An incubator was used to set up the experiment for an hour to check the effect of various temperatures (30 °C, 40 °C, 50 °C, 60 °C, 70 °C, and RT room temperature). Firstly, the synthesis of CuNP was indicated by the reaction mixture’s color change due to the reduction of Cu^++^ to Cu^0^. Then, each solution was taken in Eppendorf’s tube and centrifuged. The pellet was dissolved in distilled water and centrifuged again. The centrifugation process was repeated several times to purify the nanoparticles. Each solution was subjected to UV-spectrophotometer within the 300 nm to 800 nm wavelength to observe the spectrum. The temperature showing the maximum production of CuNP was used for the further optimization procedure.

#### 4.3.2. Effect of Incubation Time

The incubation time was optimized using 10 mL of freshly prepared root extract with 10 mL of 0.3 M CuSO_4_·5H_2_O solution in glass vials. The solutions were incubated in the dark at the selected temperature for 1, 2, 3, 4, 5, and 6 h. The change in color of the reaction mixture indicated the synthesis of CuNP. Purified CuNP was subjected to UV-spectroscopy for further confirmation of synthesis after each time interval. The incubation time showing the maximum production of CuNP was used for the further optimization procedure.

#### 4.3.3. Effect of pH 

Various pH values were considered to find the optimum value for synthesizing CuNP using the Rhatany root extract. The freshly prepared root extract was added to 0.3 M CuSO_4_·5H_2_O solution in a respective vial. The pH of the solution was maintained at 5 and kept in an incubator at the selected temperature. The color change was observed at first for the synthesis of CuNP, and then the solution was subjected to UV-spectrophotometry after an optimized time. The process was repeated at pH 7, 9, 11, and 13. NaOH and HCl were used to maintain the pH of the solutions. The pH at which the maximum yield of nanoparticles was given was selected for further procedures. 

### 4.4. Characterization of Green Synthesized CuNP

Green synthesized CuNP at optimized conditions was subjected to the following techniques to determine their size, surface structure, morphology, and other characteristics.

#### 4.4.1. UV-Spectrophotometric Analysis

This technique was used to determine the formation of CuNP. The purified CuNP was dissolved in 5 mL distilled water using a vortex and subjected to UV-vis analysis. The absorption spectrum within the wavelength range of 300–800 nm was taken on a UV-vis spectrophotometer. 

#### 4.4.2. Fourier Transform Infrared (FTIR) Analysis

Fourier transform infrared spectroscopy (FTIR) was used to study the organic functional groups attached to the surface of CuNP. The synthesized nanoparticles were purified and dried at 60 °C. The dried samples were mixed with a fine powder of potassium bromide (KBr) and analyzed by FTIR. 

#### 4.4.3. X-ray Diffraction Analysis

X-ray diffraction (XRD) was used to examine the overall oxidation state and crystal structure of CuNP.

#### 4.4.4. Electron Microscopy Analysis

Scanning electron microscopy: External morphology and surface features of CuNP was analyzed using a scanning electron microscope. The CuNP was rinsed with distilled H_2_O. Thin films of the sample were prepared on a silicon plate. A drop of CuNP was placed on a silicon plate, and then the film was allowed to dry on a hot plate at 100 °C for 2 h and then gold coated. 

Transmission electron microscopy: Crystalline nature of CuNP was characterized using TEM instrument using selected area (electron) diffraction (SAED) technique and JEOL, JEM-2100 at Central Lab, National Research Centre, Dokki, Giza, Egypt.

### 4.5. CuNP Applications

#### 4.5.1. Antioxidant Activity Test

By using the DPPH (1,1-diphenyl-2-picrylhydrazyl) radical scavenging assay (RSA), the antioxidant activity of the precursor salt (CuSO_4_·5H_2_O), extract, and bio-synthesised CuNP was evaluated [31]. To verify its stability, the DPPH radical solution was left undisturbed for three hours. The solution’s stability throughout the experiment was confirmed by the observation of a constant max of the solution at 517 nm. Subsequently, 1 mL of CuNP solutions (200, 150, 100, 50, and 25 g/mL) were combined with 1 mL of methanolic DPPH solution (0.1 mM). At 27 °C in the dark, the reaction solution was blended and incubated for 30 min. The solutions’ absorbance was measured at 517 nm with a UV-vis spectrophotometer. The positive and blank samples used were ascorbic acid. We computed the free radical scavenging activity using the following formula:$$DPPH\;scavenging\;activity\left(\%\right)=\frac{AB-AS}{AB}\times 100$$
where AB represents the blank’s absorbance and AS represents the samples.

#### 4.5.2. Antimicrobial Activity of Extract and Synthesized Nanoparticles

##### Antibacterial Activity

*Escherichia coli*, a Gram-negative bacterium, and *Staphylococcus aureus*, a Gram-positive bacterium, were used as test organisms for the antibacterial activity of extract and CuNP. Using Mueller Hinton Agar and Mulberry Hinton Broth as solid and liquid media, respectively, and in accordance with the Clinical and Laboratory Standards Institute’s (CLSI) standard methods, the antibacterial activity of the extract and Cu NPs was examined [32]. A suspension of fresh bacterial culture added to the liquid media was swabbed on a plate containing Mueller Hinton culture medium and the bacterial density was visibly adjusted to 0.5 McFarland standard reagents to achieve turbidity comparable to the McFarland standard. The disks were immersed in 50 µL of leaf extract, and 25, 50, 100, and 200 μg/mL of different concentrations of green CuNP dissolved in DMSO. Negative control, DMSO solvent and a chloramphenicol-impregnated standard disc (30 µg), were employed. To determine the bacterial strains’ susceptibility to the substances, the cultures were assessed for the inhibitory zone using ruler measurements (in mm) after being incubated at 37 °C for 24 h. Each experiment was carried out in two Petri dishes.

##### Antifungal Activity

The agar well diffusion method was used to assess the antifungal activity of extract and CuNP against *Alternaria alternata* and *Fusarium oxyporium* [33]. Sabouraud dextrose agar (SDA) culture media plates were created, inoculated with the fungus strains, and incubated at 25 °C for 10 days. After 10 days, wells were drilled into the agar plate using a cork borer, and samples at various concentrations (25, 50, 100, and 200 g/mL) were dissolved in DMSO and injected into each well. As positive and negative controls, respectively, fluconazole and DMSO were utilized. The test material was allowed to diffuse on the plates for one hour before the plates were incubated at 25 °C for three days. After incubation, the zone of inhibition against the tested fungus was assessed.

### 4.6. Statistical Analysis

Data were statistically analyzed using Origin Pro version 9.1 software. All tests were performed in triplicate. Analysis of variance (ANOVA) was performed using the least significant differences (LSD) test to compare the significant differences (*p* ≤ 0.05) between groups.

## 5. Conclusions

In this study, component phytochemicals from Rhatany root extract were used as natural reducing and capping agents for the easy and eco-friendly biosynthesis of new CuNPs. The discovery of a surface plasmon resonance (SPR) peak at 426 nm UV-visible spectroscopy verified the formation of viable CuNP in the reaction mixture. The produced nanoparticles have right optical properties for biological activity. According to the FTIR spectrum, the constituent functional groups of the extract phytochemicals were responsible for decreasing and capping biosynthesized CuNP. Spherical CuNP with average particle sizes ranging from 5.2 to 7.7 nm and face-centered cubic geometry were studied using XRD, SEM, and TEM/HRTEM-SAED. The biological activity of green copper nanoparticles produced in vitro was investigated and demonstrated remarkable ability by scavenging DPPH radicals and suppressing the growth of drug-resistant bacterial and fungal strains. Therefore, biosynthesized CuNP from Rhatany root extract could be interesting candidates for a new antimicrobial agent and antioxidant therapy.

## Figures and Tables

**Figure 1 molecules-28-04629-f001:**
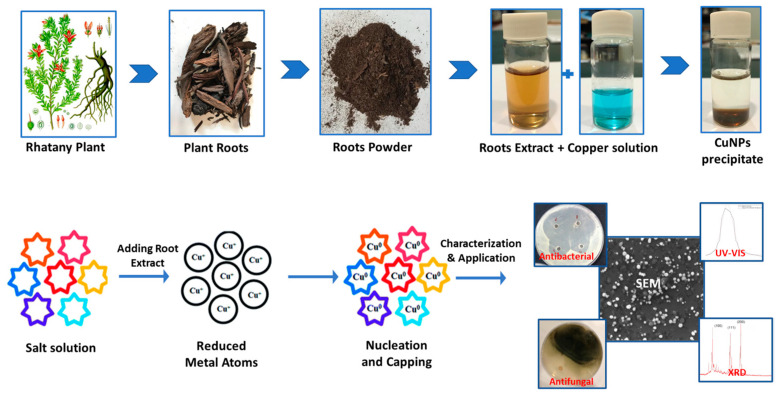
Protocol of plant Rhatay root extraction, biosynthesis mechanism of CuNP, characterization and its application.

**Figure 2 molecules-28-04629-f002:**
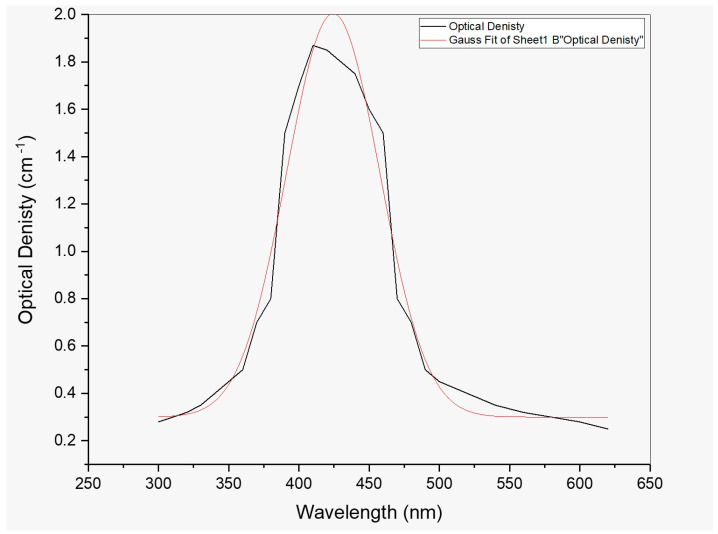
UV–Vis spectrum for Cu NPs prepared by 0.3 M precursor concentrations.

**Figure 3 molecules-28-04629-f003:**
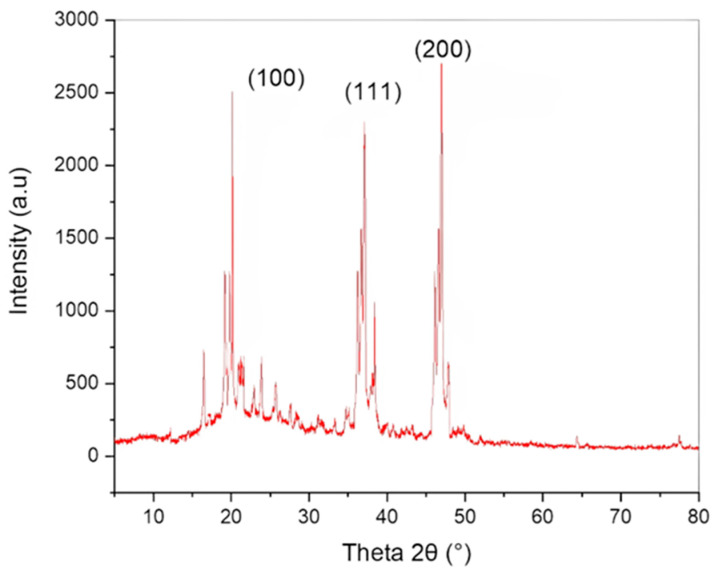
XRD pattern of biosynthesized CuNP.

**Figure 4 molecules-28-04629-f004:**
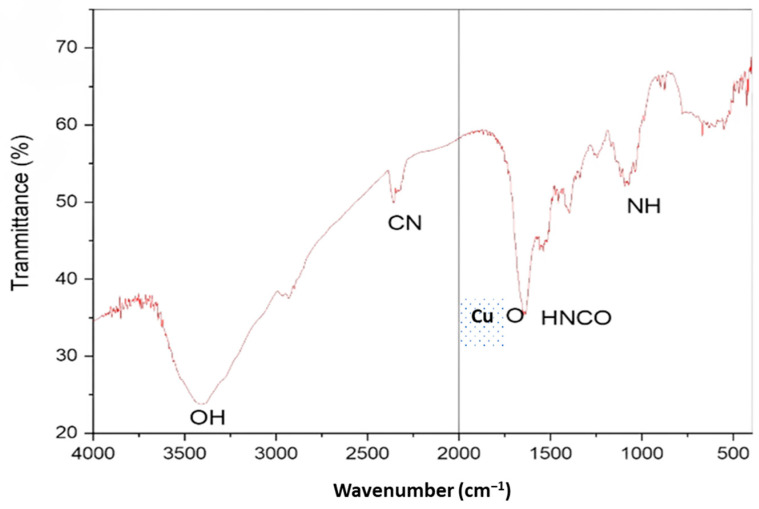
FTIR spectra of Cu NPs.

**Figure 5 molecules-28-04629-f005:**
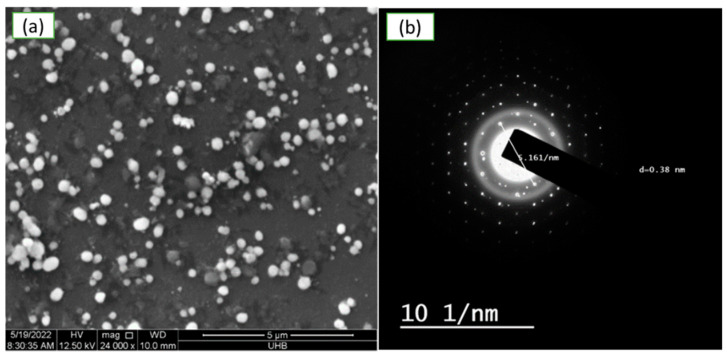
Electron micrograph of green synthesized copper nanoparticles using aqueous extract of Rhatany roots. (**a**) SEM images at 5 µm, (**b**) HRTEM pictures of the mono-distributed spherical CuNP.

**Figure 6 molecules-28-04629-f006:**
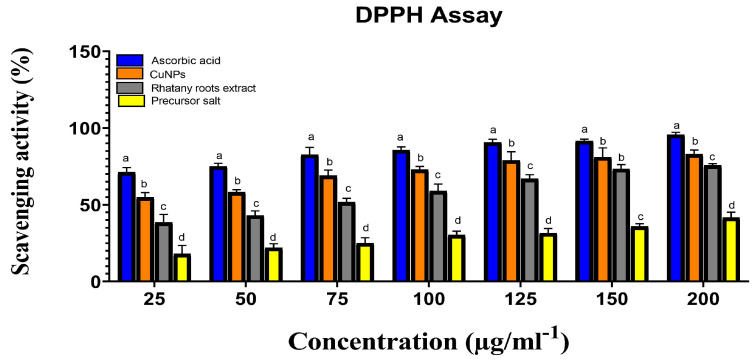
Precursor salt, Rhatany roots extract, CuNP, and ascorbic acid’s ability to scavenge DPPH radicals. Different letters indicate the significant difference at *p* > 0.05 by LSD test in each group.

**Figure 7 molecules-28-04629-f007:**
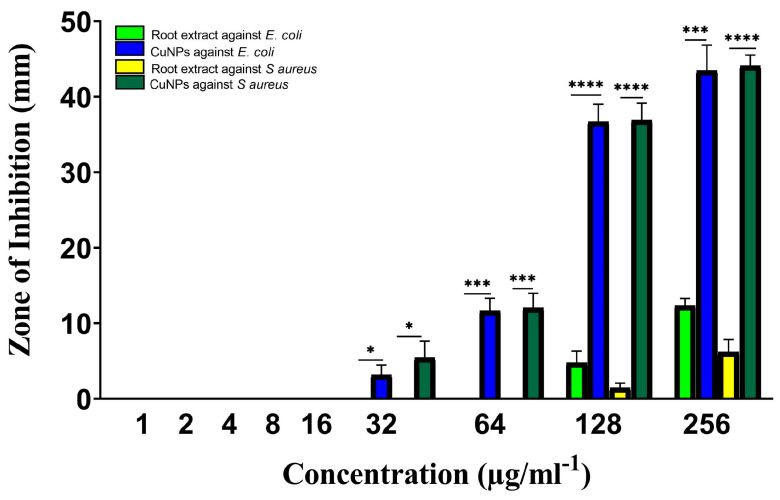
Antibacterial activity of Rhatany roots extract and CuNP against *S. aureus* and *E. coli* bacteria strains. The *p*-value: * *p* < 0.05, *** *p* < 0.001, **** *p* < 0.0001 indicated a significant correlation among different groups.

**Figure 8 molecules-28-04629-f008:**
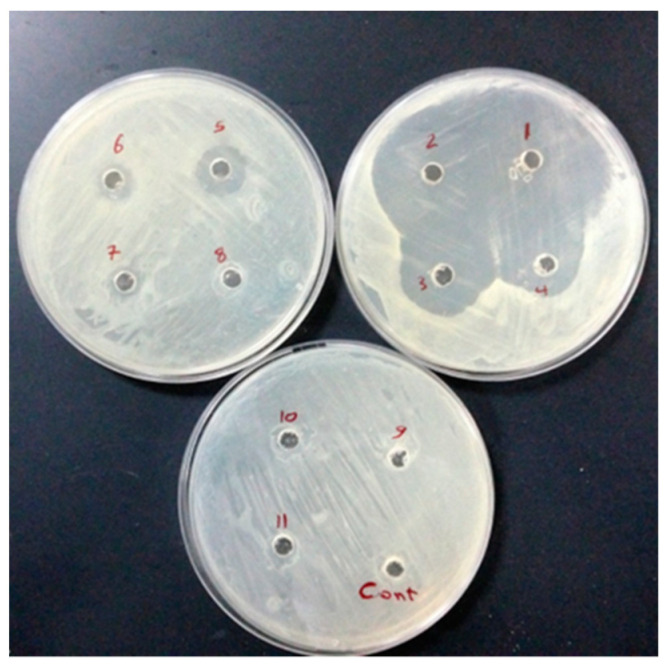
Well diffusion test was performed on 9 concentrations (1, 2, 4, 8, 16, 32, 64, 128, and 258 µg/mL^−1^) from No 11 to 3, respectively. No 1 and 2 are considered positive controls (chloramphenicol 30 µg/mL^−1^); DMSO solvent (Cont. well) is considered a negative control against *E. coli*.

**Figure 9 molecules-28-04629-f009:**
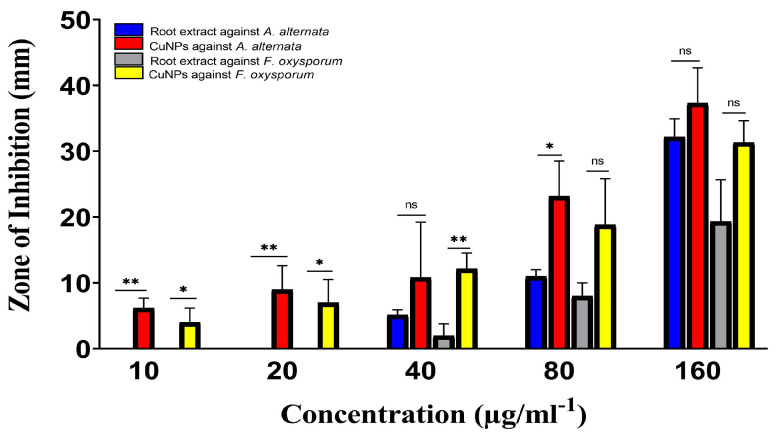
Antifungal activities of Rhatany roots extract and CuNP against *A. alternata* and *F. oxysporum* fungal strains. Data represent the mean of the zone of inhibition. The *p*-value: * *p* < 0.05, ** *p* < 0.01, indicated the significant correlation among different groups.

## Data Availability

Not applicable.

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
