# Peer review of "Synthesis of Green Copper Nanoparticles Using Medicinal Plant Krameria sp. Root Extract and Its Applications"

_molecules, 2023, doi:10.3390/molecules28124629_

Round 1

Reviewer 1 Report

The paper “Synthesis of Green Copper Nanoparticles Using Medicinal Plant Krameria sp. Root Extract and its applications”  is a research article describing a green synthesis route to obtain copper nanoparticles and their efficiency as antioxidant and antimicrobial agent. The article is suitable for publication in MDPI Molecules after minor corrections described below:

1.            Line 163 The instead of the.

2.            Line 185 change CONPs to CuNPs.

3.            Line 258 says reduction of Ag+ to Ag0. Must be corrected.

4.            Describe in the methodology the method of purification of CuNPs.

5.            One of the parameters that the authors describe from the reported literature that may influence the synthesis of CuNPs is pH. Were the pH values of their tests measured and controlled? Describe in the methodology if they did and if not, the reasons why not.

6.       How does the productivity of the green route compare to other methods of CuNPs synthesis? Is the concentration of synthesized nanoparticles higher or lower? Comment on this in the discussion.

Reviewer 2 Report

- The quality of figures 2-4 can be improved. The mentioned figures could be presented in the same way as figures 6, 7 and 9.

- Could You provide the source of equation 1?

- What about error analysis and statistics in the present work? Could You please comment Your results that way? How many samples were investigated?
